**Data Availability Statement:** All relevant data are within the paper.

# Professional competencies and public health content in the human nutrition and dietetics degree program: A qualitative study based on experts' consensus

**Panmela Soares**[1], **Carmen Vives-Cases**[1,2,3]\*, **Vicente Clemente-Gómez**[3], **Rocío Ortiz-Moncada**[1,3], **Elena Lobo-Escolar**[4,5,6], **Diego Rada-Fernández de Jauregui**[7], **Victoria Arija**[8], **Ángel R. Zapata-Moya**[9], **Mari Carmen Davó-Blanes**[1,3], **Group of participants in the Fifth Meeting of University Public Health Professors of the Degree in Human Nutrition and Dietetics**[¶]

1 Public Health Research Group, Department of Community Nursing, Preventive Medicine and Public Health and History of Science, University of Alicante, Alicante, Comunidad Valenciana, Spain, 2 CIBER in Epidemiology and Public Health (CIBERESP), Madrid, Spain, 3 Public Health Research Group, University of Alicante, Alicante, Comunidad Valenciana, Spain, 4 Department of Microbiology, Preventive Medicine and Public Health, University of Zaragoza, Zaragoza, Spain, 5 Institute of Health Research of Aragon (IIS Aragón), Zaragoza, Comunidad de Aragón, Spain, 6 Center for Network Biomedical Research in Mental Health (CIBERSAM), Ministry of Science and Innovation, Madrid, Spain, 7 Department of Preventive Medicine and Public Health, University of the Basque Country/Euskal Herriko Unibertsitatea (UPV/EHU), Vitoria-Gasteiz, Pais Vasco, Spain, 8 Nutrition and Public Health Unit, Faculty of Medicine and Health Sciences, Universitat Rovira i Virgili, Reus, Catalunya, Spain, 9 Department of Social Anthropology, Basic Psychology and Public Health, Pablo de Olavide University, Sevilla, Andalucia, Spain

¶ The complete membership of the author group is provided in the Acknowledgments.
\* carmen.vives@ua.es

## Abstract

### Objective

To gather consensus on professional competencies and basic public health content for the degree program in Human Nutrition and Dietetics (HND).

### Design

In 2018, the Fifth Meeting of University Public Health Professors took place in Zaragoza (Spain). Fourteen lecturers in the HND degree program participated from 11 Spanish universities. They identified competencies and basic content for training for the HND degree using group dynamics and consensus strategies.

### Results

The professors identified 51 basic competencies, distributed in the areas of "evaluation of population health needs" (n = 20), "development of health policies" (n = 23), and "guaranteeing provision of health care services" (n = 8). In order to reach these competencies, 35 topics were proposed organized into six thematic blocks: foundations of public health, nutritional

**Funding:** The Dr. Antoni Esteve foundation funded the Fifth Meeting of University Public Health Professors of the Degree in Diet and Human Nutrition & de publication fees of this paper. The funders had no role in study design, data collection and analysis, decision to publish, or preparation of the manuscript.

**Competing interests:** The authors have declared that no competing interests exist.

epidemiology, health problems and diet and nutrition strategies, food security, health in all policies and health promotion and education.

## Conclusion

The consensus reached serves as a reference to orient and update public health education as a part of the HND degree.

## 1. Introduction

Professional competencies in public health are those capacities needed in order to reach public health objectives such as the prevention of disease and health surveillance, protection, promotion and recuperation in individuals and groups [1]. The definition of public health shows the areas for action as well as the multi-professional nature of the field. This diversity of profiles and of activities make the definition of professional competencies difficult. Over time, different national and international initiatives have proposed defining competencies in public health. One such initiative was that proposed by the United States Institute of Medicine, which was the designation of essential functions in public health practice [2]. In order to establish a common framework in Spain in 2006, the Spanish Society for Public Health and Health Administration and the Spanish Epidemiological Society published a list of public health professional competencies [3]. This work allowed for identifying the common competencies and content of public health education for different degree programs [4, 5].

The Forum for Public Health University Professors was established in Spain in 2013. It began as a dynamic and participatory space in which to engage in debate about the professional competencies of public health and come to a consensus on the basic competencies required for different degrees as well as on appropriate curriculum to reach them [6]. In terms of forum activities, the public health competencies and content for the pharmacy degree were agreed upon in 2013 [7], for medicine in 2014 [8], veterinary medicine in 2016 [9] and nursing in 2017 [10]. With the same objective, in 2018 the forum began to focus on the Degree in Human Nutrition and Dietetics (HND).

The interest in addressing education in HND is shown by various studies carried out in different countries [11–14]. Interest is growing in the current context of increasing public health problems related to diet [15, 16]. However, HND education has been carried out differently in different countries. In Spain professional interest in nutrition as a part of public health is just beginning. It is estimated that less than two percent of nutritionists carry out their professional work in a public health context [17]. The insertion of HND in the area of public health requires the training of professionals prepared to use a public health approach to address problems related to nutrition. Thus, defining the competencies and basic content of public health education in Spain is an important point of departure to achieve this objective.

The first department of nutrition and dietetics at the university level in Europe was established in 1920, and in 1970 the European Federation for Dietician Associations was created [18]. In Spain, the diploma in HND was regulated by Royal Decree 433/1998 as first-cycle education with a three-year duration in which public health constituted a foundational course of 4.5 credits [19]. The adaptation of the diploma for the degree title was carried out in 2016 [20] taking into account the professional and specific competencies established in the recommendations *(White Books)* of the National Agency for Quality Evaluation and Accreditation (the agency linked to the Ministry of Education in Spain) [17]. However these competencies were

not established by public health professionals, nor were they established specifically for the area in question. Thus, the public health training programs in the 31 universities (15 public) where the degree was offered were highly heterogeneous. Some included contents related to the functions of public health and food policies, while others did not include these topics and put more emphasis on health planning [21, 22].

In Spain graduates of HND programs are referred to as Dietitian Nutritionist. Currently this degree program includes 240 European credits and is distributed into 4 years. The courses related to public health are required. They represent a total of 21.5 European credits that are distributed into courses such as public health, epidemiology, community nutrition and nutrition education. Furthermore, students much acquire 21.5 European credits in external practicums [17].

With respect to post-graduate training, there is no specific master or doctorate in nutrition in public health [23]. These topics make up a part of other programs such as human nutrition and dietetics, public health and health sciences.

In professional work, Dietitian/Nutritionist have difficulty finding jobs in public health. They typically work in clinics and private consultancies [17].

In order to promote the professional development of Dietitians/Nutritionists in public health, it is useful to adapt the training contents to professional competencies in public health [3, 24–26]. The objective of this study is to gather consensus on professional competencies and basic public health content for the degree program in HND.

## 2. Methods

This work comprises part of a project that has been carried out since 2013 and whose methodology has been described previously [7–10].

In this study we generated consensus on professional competencies and public health content for the HND degree through a qualitative study that used the nominal group technique [26] with professors who had experience teaching in the HND degree.

Participant selection was carried out by identifying, through the webpage of the Conference of Spanish University Rectors, those public Spanish universities (N = 50) that offered the HND degree during the 2017–18 academic year. Fifteen universities were identified distributed in 9 of the 17 Autonomous Communities.

Looking at the teaching guides of the public health courses of these universities, full-time professors who were responsible for the course were selected, preferably from the area of preventive medicine and public health. The professors in scientific areas such as food and animal science and animal production and nutrition were not included.

Contact with the professors was carried out by email. They were explained the objective and antecedents of the meeting and the reason they were being brought together. In the case they couldn't participate, a request was made for a recommendation of another individual from their area of expertise from the same university. Following the same procedure, contact was made with the suggested participant, and repeatedly this was carried out until a definitive list of participants was obtained. A total of 25 people were contacted, and 14 ended up participating in the study (10 women and 4 men) from 11 universities: University Complutense of Madrid, University of Alicante, University of Granada, University of Santiago de Compostela, University of the Basque Country, University of Zaragoza, University Pablo de Olavide, University of Barcelona, University of Lleida, University of València, and University Rovira i Virgili. Each university had a representative, except in the case of Zaragoza and Alicante, which had three and two representatives, respectively. There were five assistant professors, four associate professors, one tenured professor, one post-doctoral researcher, and three adjunct professors. Academic training of the

participants was related to a multidisciplinary public health approach. Participation in the workshop included doctors (n = 5), Dietitians/Nutritionists (n = 3), pharmacists (n = 3), nursing (n = 1), graduates in political science and administration (n = 1) and humanities graduates (n = 1). All of the participants held doctorates in public health or related areas: public health (n = 6), medicine (n = 4), pharmacy (n = 3), social sciences (n = 1). The majority of participants had more than five years of teaching experience in public health and HND: more than 10 years (n = 4), between 5 and 10 years (n = 6), less than 5 years (n = 4).

Information was collected during the December of 2018 at the Fifth Meeting of the Forum of University Public Health Professors, which took place in Zaragoza. The meeting lasted one day and a half, during which three consensus activities were carried out in groups of four or five participants and in plenary. The same professors participated in the three activities.

**The first activity** was carried out in groups of four to five participants. With the help of an online questionnaire, each group was presented with a list of 80 professional competencies in public health (defined for Spain) grouped into three essential areas: "evaluation of population health needs", "development of health policies", and "guaranteeing provision of health care services" [3]. Each group classified the competencies into one of the following categories: a. belonging to the HND degree; b. post-graduate; c. transversal (competencies that could correspond to both the degree as well as post-graduate) and; d. not applicable (considered not appropriate for the degree). The results of this first activity were presented in plenary. When less than 2/3 of the groups agreed, consensus on the competency was arrived at through consultative discussion.

**In the second activity** new groups of four to five participants were formed, different from those in the first activity. In this activity the participants discussed and agreed to a list of basic contents, which in their opinion would be necessary for development of the competencies that were agreed to in consensus. Based on their teaching programs and their own experience, they were told to agree to a maximum of five content blocks, each of which would contain a maximum of five thematic units. The content block was presented as coherent group, organized by thematic units. The thematic units also referred to the contents that would contribute to reaching the competences identified in activity 1.

**In the third activity,** the proposal of the three groups was reviewed and grouped by two experts, repeated content was excluded and differences were aggregated until a consensus of maximums was reached. The result was presented in plenary, where the number of thematic blocks and their specific titles were agreed to. Later in the plenary there was discussion about their contents until an agreement was reached. Consensus was reached through consultative discussion in which participants presented and defended their points of view. In cases of discrepancy, agreement was reached by a simple majority vote.

Prior to the meeting, we sent participants an invitation letter to inform them about the objective of our study and gather their consent to take part in it. The participation was voluntary, and all participants had the chance to review the consensus reached after the meeting ended.

## 3. Results

Of the 80 public health competencies presented to the professors, 51 were identified as basic for the HND degree, 25 for the post-graduate program, and 4 were classified as not applicable. Furthermore, the participants added a competency to the predefined list, which was agreed to by consensus as basic for the degree.

Table 1 shows the classification of the professional competencies for Dietitians/Nutritionists in the area of "evaluation of population health needs". Of the 30 competencies of this area, 25 of them obtained a high level of consensus among the groups, while the remaining 5 were

**Table 1. Competencies agreed to in the Fifth Meeting of University Public Health Professors of the Degree in Human Nutrition and Dietetics for the Area of "Evaluation of Population Health Needs".**

| Competencies (N = 30) | Degree n (%) | Post-graduate n (%) |
|---|---|---|
| **1—Analysis of the health situation of the community (n = 10)** | **9 (90)** | **1 (10)** |
| Identify factors that determine health | X | |
| Interpret health and disease processes in human populations | X | |
| Collect, process and store information (demographic and health-related) | X | |
| Evaluate the quality of the information | X | |
| Produce health indicators and indices | | X |
| Carry out quantitative and qualitative analysis of data | X | |
| Analyze the spatial and temporal distribution of health information | X[a] | |
| Evaluate the distribution of environmental, biological and cultural risks | X | |
| Evaluate health inequalities | X | |
| Produce and communicate information to diverse audiences | X | |
| **2—Describe and analyze the association and impact of risk factors on health problems and the impact on health services (n = 11)** | **6 (55)** | **5 (45)** |
| Generate a hypothesis with a scientific basis | X | |
| Design appropriate research proposals | | X[a] |
| Manage appropriate epidemiological and statistical techniques | X | |
| Design instruments to measure and collect data | | X |
| Process and store information in databases | X | |
| Evaluate data quality | | X[a] |
| Manage research techniques related to services and interventions | | X |
| Apply economic evaluation techniques | | X |
| Synthesize results appropriately | X | |
| Be familiar with pertinent bibliographical information and identify information sources, including bibliographic searches | X | |
| Respect the ethical aspects of health documentation and research | X | |
| **3 –Control diseases and emergency situations (n = 9)** | **5 (56)** | **4 (44)** |
| Verify the existence of an emergency health situation | X | |
| Analyze the causes of an emergency | X | |
| Be familiar with efficiency and safety in terms of available control measures | X | |
| Be familiar with available resources and their organization and responsibilities | X | |
| Propose extraordinary measures for the resolution of emergency situations | | X |
| Apply measures and available resources | | X[a] |
| Generate credibility and trust | X[a] | |
| Evaluate potential community reactions (risk perception) | | X[a] |
| Communicate relevant information to the population and to professionals in a crisis situation | | X |
| **TOTAL** | **20 (67)** | **10 (33)** |

[a]Competencies agreed to in the plenary session.

discussed in plenary. In the end, 20 of the 30 competencies were considered appropriate for the degree and 10 for the post-graduate program.

Table 2 shows the classification of competencies for the HND degree in the area of "development of health policies". In this area, there was consensus among the groups for 24 of the 30 competencies. The other 6 were debated in plenary until consensus was reached. In the end, 23 were classified as relevant to the degree, 6 as post-graduate and one was identified as not relevant. In

**Table 2. Competencies agreed to at the Fifth Meeting of University Public Health Professors of the Degree in Human Nutrition and Dietetics for the area of "Development of Health Policies".**

| Competencies (N = 31) | Degree n (%) | Post-Graduate n (%) |
|---|---|---|
| **1—Contribute to defining health system organization (n = 9)** | **8 (89)** | **1 (11)** |
| Use information on the health problems and health needs of the population | X | |
| Establish health priorities for a defined population | X | |
| Formulate health objectives that are comparable and measurable | X | |
| Be familiar with different health systems | X | |
| Be familiar with current health legislation and processes of development of regulations | X | |
| Be familiar with mechanisms for assignment of health resources | | X |
| Evaluate the health, economic and social impact of health policies, including inter-sectoral health policies | X | |
| Be familiar with the objectives and political priorities in the area of health | X | |
| Be familiar with European health policies and policies of international agencies and organizations | X | |
| **2—Promote the defense of health in inter-sectoral policies (n = 5)** | **3 (60)** | **2 (40)** |
| Be familiar with the objectives and priorities of public policies related to health | X | |
| Be familiar with basic environmental, work-related, agricultural, food-related, transportation, and educational legislations related to health | X | |
| Evaluate the health impact of public policies | | X |
| Negotiate the role of health in the development of public policies related to health | | X |
| Mobilize and generate public opinion in defense of health | X | |
| **3—Contribute to the design and implementation of health programs and interventions[b] (n = 12)** | **8 (67)** | **3 (25)** |
| Identify health problems, needs and inequalities in the population | X | |
| Establish health priorities for a defined population | X | |
| Interpret potential benefits and barriers to health interventions | X[a, b] | |
| Be familiar with the factors underlying human and group behavior | X | |
| Design health education programs | X | |
| Design population-based vaccination programs | NA | NA |
| Design programs for protection against environmental risks | | X[a] |
| Design food safety programs | X | |
| Design population-based secondary prevention programs | | X[a] |
| Evaluate the ethical aspects of health interventions. Design health and social care programs | | X |
| Contribute to the inter-sectoral nature of programs | X[a] | |
| Design intervention programs in the area of food and nutrition | X[c] | |
| **4—Promote social participation and strengthen the level of control of citizens over their own health (n = 5)** | **5 (100)** | **0 (0)** |
| Prepare and provide written and verbal information for diverse groups | X | |
| Facilitate and reinforce the capacity of citizens related to their own health | X | |
| Act in defense of the health of the most vulnerable groups in society | X | |
| Identify and involve community leaders in the practice of public health | X | |
| Demonstrate leadership and coordination of diverse teams | X[a] | |
| **TOTAL** | **24 (77)** | **6 (19)** |

[a]competencies agreed to in the plenary session /

[b]change in the name of the activity or competency agreed to in the plenary session /

[c]additional competency added in the plenary session / NA = *Not applicable*.

this area the participants added a competency not initially included in the list: "design of intervention programs in the area of food and nutrition", classified as relevant to the degree. For some competencies there was consensus about the modification or adaptation of the names.

Table 3 shows the classification awarded by the participations to each of the competencies of the area "guaranteeing provision of health care services". Of the 20 competencies in this area, 15 were selected by the majority of the groups and 5 of them were debated in plenary. In the end, the participants classified 8 as relevant to the degree, 9 to the post-graduate program and three as not relevant.

Based on the competencies agreed to as relevant to the degree, and after debate, 35 thematic units were grouped into six blocks: 1) fundamentals of public health, 2) nutritional epidemiology, 3) health problems and food and nutrition strategies, 4) food security, 5) health in all policies, and 6) health promotion and education. Table 4 shows the blocks and thematic units agreed to by consensus by the public health professors for the HND degree.

**Table 3. Competencies agreed to at the Fifth Meeting of University Public Health Professors of the Degree in Human Nutrition and Dietetics for the area of "Guaranteeing Provision of Health Care Services".**

| Competencies (N = 20) | Degree n (%) | Post-Graduate n (%) |
|---|---|---|
| **1—Manage services and programs (n = 5)** | **1 (20)** | **3 (60)** |
| Facilitate the accessibility of health services to vulnerable groups | NA[a] | NA[a] |
| Install health programs | | X |
| Develop budgets and prepare funding proposals | | X |
| Identify health priorities in any situation | X | |
| Manage multidisciplinary teams and resolve conflicts | | X[a] |
| **2—Evaluate services and programs (n = 4)** | **0 (0)** | **4 (100)** |
| Evaluate the efficiency, effectiveness, utility, safety, and geographic, social, ethnic or gender equity of health interventions | | X |
| Analyze population, professional and provider satisfaction with health services | | X |
| Use the methods of structure, process and results that are most appropriate in each case, including quality of life, satisfaction, acceptance, etc. | | X |
| Know how to apply established criteria for the accreditation of health services and activities | | X |
| **3—Carry out health inspections and audits (n = 6)** | **4 (67)** | **0 (0)** |
| Be familiar with current legislation on health risks | X | |
| Be familiar with action mechanisms for the primary health risks | X | |
| Be familiar with and trained in audit techniques | X[a] | |
| Be familiar with legislation applicable to each area of health regulation | X | |
| Propose and/or adopt special measures (seizures, precautionary interventions, etc.) | NA[a] | NA[a] |
| Propose and carry out action to improve service delivery | NA[a] | NA[a] |
| **4—Develop guides and protocols (n = 5)** | **3 (60)** | **2 (40)** |
| Synthesize current knowledge about the impact of health interventions of interest | X | |
| Be familiar with processes of development of guides and standardized work protocols | X | |
| Adapt available guides to certain areas | X | |
| Develop standardized control measures and procedures | | X |
| Involve relevant agents (professional associations, experts, professional representatives, etc.) in the development and application of guides and protocols | | X[a] |
| **TOTAL** | **8 (40)** | **9 (45)** |

[a]competencies agreed to in the plenary session / NA = *Not applicable.*

**Table 4. Thematic blocks and areas agreed to at the Fifth Meeting of University Public Health Professors of the Degree in Human Nutrition and Dietetics.**

**1. FOUNDATIONS OF PUBLIC HEALTH (n = 6)**

The concept of health and disease. Public health and its functions

Health determinants and inequalities

Natural history of disease and levels of prevention

Health systems. Public health legislation

Information systems and health indicators

International and national organizations related to public health

**2. NUTRITIONAL EPIDEMIOLOGY (n = 10)**

General concepts in epidemiology

Measures of frequency, association and impact

Types of study and design of studies in nutritional epidemiology

Causality

Validity and precision. Biases and random error. Diagnostic tests. Confusion

Evaluation methods for food consumption and data sources

Evaluation methods for population nutritional status

Dietary patterns

Analysis of data in nutrition studies

Evidence-based nutrition

**3. HEALTH PROBLEMS AND FOOD AND NUTRITION STRATEGIES (n = 5)**

Transmissible diseases and their prevention

Non-transmissible diseases and their prevention

Strategies and programs in public health

Nutritional recommendations and objectives. Dietary guides

Impact of information technologies in public health

**4. FOOD SECURITY (n = 5)**

Concept and indicators in food security. Food sovereignty

Food systems and sustainability

Health risks associated with food products

Health audits and inspections

Nutritional labelling

**5. HEALTH IN ALL POLICIES (n = 5)**

Healthy public policies

Interventions to reduce inequalities in health

Management and planning in health

Design of community intervention programs related to diet and nutrition and their evaluation

Dietary and nutrition policies in the European and Spanish context

**6. HEALTH PROMOTION AND EDUCATION (n = 4)**

Health promotion: concepts and strategic lines for health promotion

Health education

Design of nutrition education programs

Diet and nutrition communication strategies

## 4. Discussion

This study shows the consensus reached by a subset of 14 professors of the 11 public Spanish universities that teach in the degree in HND, with respect to the professional competencies and basic content areas of public health that should be included in the degree program. The competencies that were agreed to by consensus correspond to three areas of public health,

with a greater number of competencies for the area of "development of health policies", followed by "evaluation of population health needs", and finally "guaranteeing provision of health care services". To achieve these competencies, participants proposed 35 thematic units related to public health and epidemiology, public health policy, strategies for the prevention of poor health related to diet and nutrition, health promotion and food security.

Our results identify a series of professional competencies in public health that any HND student should develop during the degree program. However, they do not include all the basic competencies defined by the Spanish Public Health Society for professionals who work in public health. These results are similar to those shown in prior studies for other degree programs in Spain [7–10] and could be explained by the diversity of activities carried out in the area of public health and by the multidisciplinary nature of the field.

Despite the efforts of different organizations to define areas of action and professional competencies for Dietitians/Nutritionists [27], their activity in public health is just beginning. This could be explained by the fact that the degree in HND is relatively recent, and the inclusion of these professionals in the health system is not yet consolidated in the majority of countries in Europe. These results warrant recognition, because the number of competencies in public health attributed to this type of professional is greater than what has been proposed for other degrees in prior studies [8, 10]. This shows the great potential that participants attribute to Dietitians/Nutritionists in carrying out activities related to prevention, protection, surveillance and promotion of population health.

Similar to the consensus achieved for the degrees in nursing and medicine in Spain [8, 10], the number of competencies identified for the area of "guaranteeing provision of health care services" was lower. This was the case for the competencies related to the management and evaluation of services and programs, which agrees with other prior studies in other countries [13, 28], considered appropriate for post-graduate programs. Thus, our findings show the relevance of post-graduate programs for achieving development of these competencies in nutrition and dietetics. In this way, ethics, communication and leadership are included in various competencies, which, according to our results, should be initiated during the degree and completed during post graduate. In fact, there is evidence that professionals with a postgraduate degree feel that they have greater abilities in terms of planning and administrative evaluation, as well as in terms of organization, research, oversight and leadership [11].

In terms of content, the consensus of HND proposed an initial module to approach general aspects of public health, such as the concept of health and its determinants. Also, content areas were established for epidemiology and health problems. Furthermore, a module was proposed with specific topics from the profession. The study of the state of nutrition was proposed and the food consumption of the population as well as to give greater emphasis to health problems related to diet. These results show a common theme in training in public health for Dietitians/Nutritionists and other health professions, which should be complemented by topics that are specific to each profession.

Health promotion was considered an important topic, and a specific block was agreed to in order to address the issue. On the contrary, contents related to surveillance received less attention, possibly because this theme has been attributed generally to other professions. However, the increase in health problems related to diet [15] shows the need to strengthen a dietary and nutritional surveillance system that supports the process of evaluation and planning of diet and nutrition strategies. This topic could be taught in the HND degree, taking into account that surveillance is one of the action areas of the World Health Organization World Strategy on Diet, Physical Activity and Health [29].

In addition to the consensus in the areas of veterinary health and pharmacy [7, 9], a thematic block was established titled "food security" for the HND degree, albeit with different

content. While in veterinary health and pharmacy the contents are focused on hygiene-health aspects, nutrition incorporates other dimensions, such as the availability and access to food, the stability of food production and the sustainability of the food system [30, 31].

Overall, the thematic blocks proposed in our study are similar to contents included in the HND degree in other countries [32–35]. However, each university distributes the topics into different courses. While in Spanish universities the topics are taught in public health, community health, epidemiology and nutrition education courses [21, 22], other countries provide courses such as "social determinants", "food security", "food systems" and "health systems" [32–35].

In interpreting these results, it should be taken into account that the list of competencies used as a reference in our study establishes a common base for the actions of different public health professionals and does not distinguish between area of influence [14]. This makes it difficult to compare this study with other studies that have identified specific competencies in public health for Dietitians/Nutritionists [11–14]. However, the methodology used allowed us to identify those competencies (from among the general competencies in public health) that are considered specific for the HND degree, based on the opinions of experts in the field of nutrition and public health.

On the other hand, the participation of professors from different universities with long histories in public health represents an opportunity to contrast and debate contents that are traditionally included in public health courses for the HND degree, which could help introduce changes into the contents addressed.

Certainly, the consensus achieved constitutes an important point of departure to orient training in public health in the HND degree. The definition of content in public health for the training of Dietitians/Nutritionists is especially important given the changes in the food system and in the epidemiological profile of the population. The new demands that generate these changes for public health should be incorporated into the training of graduates of HND. The proposal of content could orient and update the training of Dietitians/Nutritionists in public health, helping them to be more prepared to address topics related to food and nutrition from a public health perspective.

## Acknowledgments

To the Spanish Society of Epidemiology, the Spanish Society for Public Health and Health Administration. To the Faculty of Health and Sports Sciences at the University of Zaragoza for its support in the organization of Fifth Meeting of University Public Health Professors of the Degree in Human Nutrition and Dietetics and to the participants of said meeting. **Participants in the Fifth Meeting of University Public Health Professors of the Degree in** Human Nutrition and Dietetics: Arribas Monzón, Federico (Microbiology, Preventive Medicine and Public Health, University of Zaragoza, Zaragoza), Briones-Vozmediano, Erica (Nursing and Physiotherapy, University of Lleida, Lleida), Calle Purón, María Elisa (Public Health and Maternal and Infant Health, Universidad Complutense of Madrid, Madrid), Grau, María (Medicine, University ofBarcelona, Barcelona), Moreno, Belén (Microbiology, Preventive Medicine and Public Health, University of Zaragoza, Zaragoza), Regueira Méndez Carlos (Psychiatry, Radiology, Public Health, Nursing and Medicine, University of Santiago de Compostela, Compostela), Salcedo Bellido Inmaculada (Preventive Medicine and Public Health, University of Granada, Granada).

## Author Contributions

**Conceptualization:** Carmen Vives-Cases, Mari Carmen Davó-Blanes.

**Data curation:** Panmela Soares, Vicente Clemente-Gómez.

**Formal analysis:** Panmela Soares, Carmen Vives-Cases, Mari Carmen Davó-Blanes.

**Funding acquisition:** Carmen Vives-Cases, Mari Carmen Davó-Blanes.

**Investigation:** Panmela Soares, Carmen Vives-Cases, Vicente Clemente-Gómez, Rocío Ortiz-Moncada, Elena Lobo-Escolar, Diego Rada-Fernández de Jauregui, Victoria Arija, Ángel R. Zapata-Moya, Mari Carmen Davó-Blanes.

**Methodology:** Carmen Vives-Cases, Mari Carmen Davó-Blanes.

**Project administration:** Carmen Vives-Cases, Mari Carmen Davó-Blanes.

**Supervision:** Carmen Vives-Cases, Mari Carmen Davó-Blanes.

**Writing – original draft:** Panmela Soares, Vicente Clemente-Gómez.

**Writing – review & editing:** Carmen Vives-Cases, Rocío Ortiz-Moncada, Elena Lobo-Escolar, Diego Rada-Fernández de Jauregui, Victoria Arija, Ángel R. Zapata-Moya, Mari Carmen Davó-Blanes.

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
