## [Decision Letter · Decision Letter 0]

3 Nov 2020

PONE-D-20-13991

Original Articles: Professional Competencies and Public Health Content in the Diet and Human Nutrition Degree Program: a qualitative study based on experts' consensus

PLOS ONE

Dear Dr. Vives-Cases,

Thank you for submitting your manuscript to PLOS ONE. After careful consideration, we feel that it has merit but does not fully meet PLOS ONE’s publication criteria as it currently stands. Therefore, we invite you to submit a revised version of the manuscript that addresses the points raised during the review process.

We look forward to receiving your revised manuscript.

Kind regards,

Mauro Lombardo

Academic Editor

PLOS ONE

Journal Requirements:

2. One of the noted authors is a group; Grupo de la V reunión del foro de profesorado Universitario de Salud Pública en el Grado de Nutrición Humana y Dietética.

In addition to naming the author group, please list the individual authors and affiliations within this group in the acknowledgments section of your manuscript.

Please also indicate clearly a lead author for this group along with a contact email address.

Reviewers' comments:

Reviewer's Responses to Questions

**Comments to the Author**

1. Is the manuscript technically sound, and do the data support the conclusions?

Reviewer #1: No

Reviewer #2: Yes

Reviewer #3: No

Reviewer #4: Yes

2. Has the statistical analysis been performed appropriately and rigorously? 

Reviewer #1: N/A

Reviewer #2: Yes

Reviewer #3: N/A

Reviewer #4: N/A

3. Have the authors made all data underlying the findings in their manuscript fully available?

Reviewer #1: No

Reviewer #2: Yes

Reviewer #3: Yes

Reviewer #4: No

4. Is the manuscript presented in an intelligible fashion and written in standard English?

Reviewer #1: Yes

Reviewer #2: Yes

Reviewer #3: No

Reviewer #4: Yes

5. Review Comments to the Author

Reviewer #1: The manuscript presents the interesting and well described study, however it should not be published in this high IF journal. In my opinion the scientific value of this work is rather low. The references include not current literature from low impacted journal or other sources.

Reviewer #2: I was honored to review the manuscript entitled “Professional Competencies and Public Health Content in the Diet and Human Nutrition Degree Program: a qualitative study based on experts' consensus” submitted to Plos One. The study presents high quality and deals with important clinical issue, such type of study is needed. I have only few small remarks that authors should address properly.

I recommend to accept the manuscript after minor revision.

There are only some points to correct:

- please provide the list of abbreviations

- please provide the number of ethical approval

- introduction and discussion section need improvement – please provide information on how your results will translate into clinical practice

- in discussion section please provide study strong points and study limitation section

- please correct typos

I recommend to accept the manuscript after minor revision.

Reviewer #3: In this manuscript, the authors describe a consensus in the classification of professional competencies and basic public health content (already stated in Spain) to orientate the degree in Human Nutrition and Dietetic. In the consensus participated 14 lecturers from 11 Spanish universities which are part of the degree program of DHN. Results show competencies and basic content for training for the DHN. Author conclude that the consensus may be helpful as a reference to orient and update public health.

General: Overall, it is an interesting topic and very necessary to look into. The paper needs better clarity in several aspects. The introduction, author should give a clear and detailed background which will set the reader in the context, mainly because the nutritionist or dietitian profession is differently regulated in some countries and, and has different levels of development in the countries. In addition, aspects of the methodology need more detail, as how the consensus was reach, did the methodology included any instrument (validated)? The discussion, authors present their finding but the paper would be more interesting to read if there would be an analysis of what competencies actually imply in DHN programs.

Specific aspects:

1. The full proposed title is: “Professional Competencies and Public Health Content in the Diet and Human Nutrition Degree Program: a qualitative study based on experts' consensus”. However, in reference 19, 21 and 22 the degree program is “Diplomado en Nutrición Humana y Dietética”, it seems to me that DHN (referred by authors as Degree in Diet and Human Nutrition) should be Degree in Human Nutrition or Human Nutrition and Dietetic. This should be corrected in several parts of the article.

2. Some aspect are confusing or need to be clarified in the introduction. According to the article, Degree DHN forms Dietitians or Nutritionists, along the article, mainly in the discussion the term “dietary nutritionists” is mentioned. Is that a different term for referring a Dietitian or Nutritionist working in public health? Is a formal term?.

3. Because Dietitian/Nutritionist is regulated in different ways in different countries, I suggest to include on Introduction some local aspect of the profession like the existing areas in which Dietitian/Nutritionist are insert now (this is particular pertinent because later on the results some topics as Health risks associated with food products and Health audits and inspections are mentioned and those are mainly part of other areas related to manage of food system, staff and budget to provide safe and nutritious meals at hospitals/long term residences/collective feeding or Food services. Moreover, how it has been the profession´s development in the last years. It would be interesting to see a background of the DHN and its role in public health so far.

4. Regarding Methodology,

1) References 7-10 gives a detailed of the steps, however more detail is needed in how the consensus was reached (Delphi panel, or other? Any validated instrument). Did authors checked/included any international experience or literature to know the role of Dietitian/Nutritionist in public health?.

2) It would be important to clarify in details some characteristics of these group of 14 lecturers. Although is mentioned the categories of each of them (line 138) and also that professors had an experience in public health, is not clear if they were physicians, nurse, dietitian, etc. nor if any of them had a PhD in Nutrition Public Health?. I think is important to state if participants were highly qualified and with experience in public health related to food/ nutrition/disease/promotion/food safety/community nutrition/community education.

3) Additionally, and very important, although competencies related to public health may be common to all public health practitioners; is necessary to describe discipline-specific competencies for future dietitians/nutritionist professionals (discipline-specific competencies provide unique or technical skills, knowledge and abilities). Table 4 intent to do it but methodically is not clear how these competencies came up.

4. Regarding the 3 areas and list of 80 professional competencies in public health, defined for Spain. I have one comments. How different universities did define competencies in their respective programs? Was in this consensus any evaluation about how we are and we are we going?.

5. Regarding results section, it should be more analysis of generic and specific competencies, to emphasize what corresponds to health professionals and what to dietitian/nutritionist. A critical analysis of the literature or a state of the art of the current is required in order to identify generic and specific competencies in a DHN program. One of the most important objectives of the competencies is to discriminate relevant aspects from the general ones, to structure a curricular sequence of the contents in the programs.

6. Finally, It is important to discuss what is the importance of competencies like leadership, management, professionalism, communication and, ethics in the Nutritionist professionals. Are any of those considered? .

Other comments

- Line 124: when authors say “preferably from the area of preventive medicine and public health” which were the other areas that not were included?.

- Table 3, number 2: “Evaluation methods for population nutrition status”, should say “nutritional status”. Same for Analysis of data in nutrition studies (nutritional studies). Number 3: “Health problems and Dietary and Nutrition Strategies” (is should be nutritional strategies.

- The abstract mentions (line 57): “Fourteen lecturers in the degree program of DHN participated from 11 Spanish universities”; line 132 “14 ended up participating in the study (10 women and 4 men) from 11 universities” and line 198 “of 11 professors of the 15 public Spanish universities”.

- It is not clear the expression “ill-health prevention strategies..” (line 204).

Reviewer #4: This article aimed to identify through consensus, professional competencies and basic public health content for the degree programmes in Diet and Human Nutrition (DHN) offered in Spain. This is an important aspect of curriculum planning and helps align the competencies of graduates with the nation’s needs. Overall, the article is well written, and the authors are commended for it. It was a pleasure to review the article.

Some minor edits are suggested to improve the readability of the manuscript, especially to an international audience.

Improving Clarity for an International Audience

For international readers, there is a need to define the scope of the “dietitians” and “nutritionists” in the context of Spain. In some countries, nutritionists deal with public health and preventive aspects of population health and dietitians with the curative aspects. Hence defining the role of “dietary nutritionists” in Spain in the introductory paragraph will be useful.

Secondly, can the authors define thematic blocks and areas and units clearly? Educations in different continents use these terms differently. In the second activity was there any specific reason to restrict the number of “blocks “to 5 and the number of units in each block to “5”. Were these limitations adhered to in the final decision? Were these ideations led by the organization of the system of delivery of DHN degrees? In Spain, are undergraduate degrees usually of 3 or 4 years duration? Is specialization possible in the undergraduate or postgraduate level? For instance, can students choose to do clinical nutrition and forego some aspects of public health? Introducing these aspects will better help readers understand the context of identifying and organizing these competencies.

In line 93, the authors state as follows- “The interest in addressing education in DHN is shown by various studies carried out in different countries [11-14] and is growing in a context of increasing public health problems related to diet”. Furthermore, in line 253- the authors also say that the competencies identified through consensus in this initiative are also similar to those identified in other countries (32-25). Since the authors allude to the exercise having been completed in other countries, it would be important for the authors to justify the need for the exercise to be done specifically for Spain. This would be especially so when they quote that a similar exercise has been completed for Europe.

Also, in the discussion session, the authors outline similarities in competencies identified with other countries (eg Australia). If there are similarities, could it be linked to the similarities in their health care systems and the role of the nutritionist/dietitian among these countries? Do these countries face similar nutritional issues? I would foresee the public health content would be slightly different for countries predominantly dealing with undernutrition and related problems versus those dealing with obesity and related risks. Countries suffering the double burden would probably have other demands too.

Line 104- The authors record that there is some heterogeneity in the public health content offered to DHN degree programmes currently offered across Spain. Can the authors give a few examples for the readers to understand the extent of this heterogeneity and the themes that commonly differed between universities?

Why were current practitioners not consulted?

Other minor suggestions

Line 80 “One of these initiatives is the Institute of Medicine of the United States, which has proposed 81 essential functions [2].”- The sentence needs to be rephrased for clarity and completeness.

Line 119- requires to be rephrased for clarity.

Line 141- Information was collected during the 19 and 20 of December- say 19th and 20th. Is there a need to mention the specific dates?

Somewhere before line 145, there is a need to say that the same 14 participants participated in all three activities.

Line 152- needs to be rephrased for clarity. Does this sentence mean that when less than 2/3 groups agreed, consensus on the competency was arrived at through consultative discussion?

6. PLOS authors have the option to publish the peer review history of their article (what does this mean?). If published, this will include your full peer review and any attached files.

Reviewer #1: No

Reviewer #2: No

Reviewer #3: No

Reviewer #4: No

---

## [Author Response · Author response to Decision Letter 0]

15 Dec 2020

Reviewer #1: The manuscript presents the interesting and well described study, however it should not be published in this high IF journal. In my opinion the scientific value of this work is rather low. The references include not current literature from low impacted journal or other sources.

Response: We appreciated your time in reviewing this work. In this new version we have incorporated the suggestions of the reviewers, and we believe that this has contributed to improvements in the manuscript. 

Reviewer #2: I was honored to review the manuscript entitled “Professional Competencies and Public Health Content in the Diet and Human Nutrition Degree Program: a qualitative study based on experts' consensus” submitted to Plos One. The study presents high quality and deals with important clinical issue, such type of study is needed. I have only few small remarks that authors should address properly. I recommend to accept the manuscript after minor revision

Response: We appreciate your comments and the opportunity to review this manuscript. 

There are only some points to correct:

1. please provide the list of abbreviations 

Response: We have reviewed the manuscript to ensure that the abbreviations used are correctly described. Furthermore, we have eliminated those that are not essential and left only those related to Human Nutrition and Dietetics (HND).

2. please provide the number of ethical approval

Response: Previous to the meeting, we sent participants an invitation letter to inform them about the objective of our study and obtain their consent to take part in it. Given that participation was voluntary and all participants had the chance to review the consensus reached after the meeting ended, we did not consider it necessary to ask for the approval of our university ethical committee. 

We included this information in the methodology section. Please see the last paragraph in the methodology section. 

3. introduction and discussion section need improvement – please provide information on how your results will translate into clinical practice

Response: In this new version we have included a brief reference to the way in which the study results can be transferred to clinical practice. Specifically, we highlight that defining the competencies and contents in public health is a point of departure in training professionals capable of addressing public health problems related to nutrition and dietetics. Please see the third paragraph of the introduction and the final paragraph of the discussion section. 

4. in discussion section please provide study strong points and study limitation section 

Response: The participation of professors from different universities and who have different professional trajectories in public health represents an opportunity to debate new educational proposals that bring up to date the contents traditionally laid out in teaching guides for this material. In response to the comment provided by the reviewer, we have included this information in the discussion. Please see the tenth paragraph of the discussion. 

5. Please correct typos

Response: We have reviewed the text and made all needed corrections. 

I recommend to accept the manuscript after minor revision.

Reviewer #3: In this manuscript, the authors describe a consensus in the classification of professional competencies and basic public health content (already stated in Spain) to orientate the degree in Human Nutrition and Dietetic. In the consensus participated 14 lecturers from 11 Spanish universities which are part of the degree program of DHN. Results show competencies and basic content for training for the DHN. Author conclude that the consensus may be helpful as a reference to orient and update public health.

General: Overall, it is an interesting topic and very necessary to look into. The paper needs better clarity in several aspects. The introduction, author should give a clear and detailed background which will set the reader in the context, mainly because the nutritionist or dietitian profession is differently regulated in some countries and, and has different levels of development in the countries. In addition, aspects of the methodology need more detail, as how the consensus was reach, did the methodology included any instrument (validated)? The discussion, authors present their finding but the paper would be more interesting to read if there would be an analysis of what competencies actually imply in DHN programs.

Response: We appreciate the comments and the opportunity to review the manuscript. In this new version we have incorporated your suggestions, of which we provide a detailed description here. 

Specific aspects:

1. The full proposed title is: “Professional Competencies and Public Health Content in the Diet and Human Nutrition Degree Program: a qualitative study based on experts' consensus”. However, in reference 19, 21 and 22 the degree program is “Diplomado en Nutrición Humana y Dietética”, it seems to me that DHN (referred by authors as Degree in Diet and Human Nutrition) should be Degree in Human Nutrition or Human Nutrition and Dietetic. This should be corrected in several parts of the article.

Response: We agree with the reviewer. The term in the text has been reviewed and homogenized across the text as Human Nutrition and Dietetics. 

2. Some aspect are confusing or need to be clarified in the introduction. According to the article, Degree DHN forms Dietitians or Nutritionists, along the article, mainly in the discussion the term “dietary nutritionists” is mentioned. Is that a different term for referring a Dietitian or Nutritionist working in public health? Is a formal term?

Response: In Spain these professionals are referred to as Dietitian Nutritionist. We have made this aspect more clear in the introduction and homogenized the term in the text. Please see the fifth and seventh paragraph of the introduction. 

3. Because Dietitian/Nutritionist is regulated in different ways in different countries, I suggest to include on Introduction some local aspect of the profession like the existing areas in which Dietitian/Nutritionist are insert now (this is particular pertinent because later on the results some topics as Health risks associated with food products and Health audits and inspections are mentioned and those are mainly part of other areas related to manage of food system, staff and budget to provide safe and nutritious meals at hospitals/long term residences/collective feeding or Food services. Moreover, how it has been the profession´s development in the last years. It would be interesting to see a background of the DHN and its role in public health so far.

Response: We have included information on the Dietitian nutritionist profession in Spain and its role in public health. Please see the fourth paragraph of the introduction. 

4. Regarding Methodology,

1) References 7-10 gives a detailed of the steps, however more detail is needed in how the consensus was reached (Delphi panel, or other? Any validated instrument). Did authors checked/included any international experience or literature to know the role of Dietitian/Nutritionist in public health?

Response: In order to carry out this study, we used a list of public health competencies produced as a part of a prior study for the Spanish context. These competencies were defined taking into account the health, social and political situation of the country. They are competencies that public health professionals should develop, independently of the area in which they carry out their professional activities. Therefore, in this study we did not include competencies specific to dieticians/nutritionists identified in other countries. In our study the objective was to identify which of the public health competencies defined for Spain are those relevant for dieticians/nutritionists. 

Consensus was carried out in two phases. I the first, an agreement was reached about which competencies should be included in the degree program. Then, in the second phase, there was discussion about which contents should be included to reach said competencies. In this new version we have included more information in the methodology to explain this consensus process. Please see the seventh and ninth paragraph of the methodology section. 

2) It would be important to clarify in details some characteristics of these group of 14 lecturers. Although is mentioned the categories of each of them (line 138) and also that professors had an experience in public health, is not clear if they were physicians, nurse, dietitian, etc. nor if any of them had a PhD in Nutrition Public Health?. I think is important to state if participants were highly qualified and with experience in public health related to food/ nutrition/disease/promotion/food safety/community nutrition/community education.

Response: We have included more information on the study participants. Please see the fifth paragraph of the methodology. 

3) Additionally, and very important, although competencies related to public health may be common to all public health practitioners; is necessary to describe discipline-specific competencies for future dietitians/nutritionist professionals (discipline-specific competencies provide unique or technical skills, knowledge and abilities). Table 4 intent to do it but methodically is not clear how these competencies came up.

Response: We agree with your comments. However, as we state in the introduction, the specific degree competencies are laid out in the white papers related to the degree. In any case, degree competencies were not always established for professionals in public health. In fact, one of the strengths of this study is its establishment of basic public health competencies for the degree from the perspective of faculty who are experts in this area. 

On the other hand, what is shown in Table 4 is not competencies, rather Table 4 shows the contents that were agreed upon in the workshop to establish public health competencies that are considered basic for the degree. These are shown in Tables 1, 2 and 3. 

4. Regarding the 3 areas and list of 80 professional competencies in public health, defined for Spain. I have one comments. How different universities did define competencies in their respective programs? Was in this consensus any evaluation about how we are and we are we going?

Response: The competencies that served as references for the development of programs were the specific competencies established in the recomendations (white books) produced by the National Agency for Evaluation of Quality and Accreditation (the agency linked to the Ministry of Education in Spain). However, these competencies are not exclusively related to public health. In this new version, we have included more information on this aspect of the introduction. Please see the fourth paragraph of the introduccion. 

5. Regarding results section, it should be more analysis of generic and specific competencies, to emphasize what corresponds to health professionals and what to dietitian/nutritionist. A critical analysis of the literature or a state of the art of the current is required in order to identify generic and specific competencies in a DHN program. One of the most important objectives of the competencies is to discriminate relevant aspects from the general ones, to structure a curricular sequence of the contents in the programs.

Response: The strength of this study is that it extracts from the general public health competency proposal those competences considered specific for the degree in Human Nutrition and Dietetics based on the opinion of experts with experience in nutrition and public health. We have included this information in the discussion. Please see the Ninth paragraph of the discussion. 

6. Finally, It is important to discuss what is the importance of competencies like leadership, Hemmanagement, professionalism, communication and, ethics in the Nutritionist professionals. Are any of those considered?

Response: In this new version, we have included in the discussion the importance of competencies such as leadership, management, communication and ethics of professional dieticians/nutritionists. Please see the fourth paragraph of the discussion. 

Other comments/

- Line 124: when authors say “preferably from the area of preventive medicine and public health” which were the other areas that not were included?

Response: In Spain courses may be shared between professors from different areas. During the participant selection process preference was given to those working in the area of preventive medicine and public health. Some of the areas excluded were food science and animal production. In this new version we have included this information. See fourth paragraph of the methodology.

- Table 3, number 2: “Evaluation methods for population nutrition status”, should say “nutritional status”. Same for Analysis of data in nutrition studies (nutritional studies). Number 3: “Health problems and Dietary and Nutrition Strategies” (is should be nutritional strategies.

Response: Thank you for the review. The text has been corrected. See Table 4. 

- The abstract mentions (line 57): “Fourteen lecturers in the degree program of DHN participated from 11 Spanish universities”; line 132 “14 ended up participating in the study (10 women and 4 men) from 11 universities” and line 198 “of 11 professors of the 15 public Spanish universities”.

Response: Thank you for the review. The text has been corrected. Please see the first paragraph of the discussion. 

- It is not clear the expression “ill-health prevention strategies..” (line 204).

Response: Thank you for the correction. We have changed the expression. Please see the first paragraph of the discussion. 

Reviewer #4: This article aimed to identify through consensus, professional competencies and basic public health content for the degree programmes in Diet and Human Nutrition (DHN) offered in Spain. This is an important aspect of curriculum planning and helps align the competencies of graduates with the nation’s needs. Overall, the article is well written, and the authors are commended for it. It was a pleasure to review the article. Some minor edits are suggested to improve the readability of the manuscript, especially to an international audience. Improving Clarity for an International Audience

Response: We thank you for the comments and the opportunity to revise the manuscript. 

1. For international readers, there is a need to define the scope of the “dietitians” and “nutritionists” in the context of Spain. In some countries, nutritionists deal with public health and preventive aspects of population health and dietitians with the curative aspects. Hence defining the role of “dietary nutritionists” in Spain in the introductory paragraph will be useful.

Response: In this new version we have included information on the dietician/nutritionist profession in Spain and its role in public health. Please see the fifth and seventh paragraph of the introduction. 

1. Secondly, can the authors define thematic blocks and areas and units clearly? Educations in different continents use these terms differently. 

Response: In this new version we have included in the methodology the definition of “blocks” and “thematic units”. Please see the eighth paragraph of the methodology. 

2. In the second activity was there any specific reason to restrict the number of “blocks “to 5 and the number of units in each block to “5”. Were these limitations adhered to in the final decision? 

Response: In activity 2 we opted to define a maximum number of blocks and thematic units to support viable discussion and in order to reach an agreement. In activity 3, the number of blocks and units was not restricted in order to reach a maximum agreement. We have included this information in the methodology. Please see the ninth paragraph of the methodology section. 

3. Were these ideations led by the organization of the system of delivery of DHN degrees? 

Response: The working groups carried out activities in an autonomous way based on the programs and teaching experience. In this new version we have included this information in the methodology. Please see the eighth paragraph of the methodology. 

4. In Spain, are undergraduate degrees usually of 3 or 4 years duration? Is specialization possible in the undergraduate or postgraduate level? For instance, can students choose to do clinical nutrition and forego some aspects of public health? Introducing these aspects will better help readers understand the context of identifying and organizing these competencies.

Response: We have included information in the introduction about educational requirements for dieticians/nutritionists in Spain. Please see the fifth and sixth paragraph of the introduction. 

3. In line 93, the authors state as follows- “The interest in addressing education in DHN is shown by various studies carried out in different countries [11-14] and is growing in a context of increasing public health problems related to diet”. Furthermore, in line 253- the authors also say that the competencies identified through consensus in this initiative are also similar to those identified in other countries (32-25). Since the authors allude to the exercise having been completed in other countries, it would be important for the authors to justify the need for the exercise to be done specifically for Spain. This would be especially so when they quote that a similar exercise has been completed for Europe

Response: In response to the comments of the reviewer, we have included information in the introduction that justifies carrying out this study in Spain. Please see the third paragraph of the introduction. 

4. Also, in the discussion session, the authors outline similarities in competencies identified with other countries (eg Australia). If there are similarities, could it be linked to the similarities in their health care systems and the role of the nutritionist/dietitian among these countries? Do these countries face similar nutritional issues? I would foresee the public health content would be slightly different for countries predominantly dealing with undernutrition and related problems versus those dealing with obesity and related risks. Countries suffering the double burden would probably have other demands too. 

Response: We agree with the fact that the needs of the countries could be quite different. However, the problems of obesity and undernutrition require a similar approach from a general public health perspective. This means that the public health competencies needed by professional dieticians/nutritionists are the same. Despite this, what could be different is the importance given to each competency as well as the weight of the content areas proposed to address them. 

5. Line 104- The authors record that there is some heterogeneity in the public health content offered to DHN degree programmes currently offered across Spain. Can the authors give a few examples for the readers to understand the extent of this heterogeneity and the themes that commonly differed between universities?

Response: Some programs include content on the functions of public health and food policies, while others do not include these areas, rather they emphasize health planning. This information has been incorporated into the introduction. Please see the fourth paragraph of the introduction. 

6. Why were current practitioners not consulted?

Response: We consider that the opinions of health professionals is very relevant. However, given that the aim of the workshop was to update teaching guides for the degree program, we opted to focus debate and discussion among the faculty who were experts in public health as a point of departure to widen the debate in future discussions with health professionals. 

Other minor suggestions

Line 80 “One of these initiatives is the Institute of Medicine of the United States, which has proposed 81 essential functions [2].”- The sentence needs to be rephrased for clarity and completeness.

Response: We have rewritten the phrase. 

Line 119- requires to be rephrased for clarity.

Response: The phrase has been revised. 

Line 141- Information was collected during the 19 and 20 of December- say 19th and 20th. Is there a need to mention the specific dates?

Response: We have rewritten this section. Please see the sixth paragraph of the methodology. 

Somewhere before line 145, there is a need to say that the same 14 participants participated in all three activities.

Response: We have included the information. Please see the sixth paragraph of the methodology. 

Line 152- needs to be rephrased for clarity. Does this sentence mean that when less than 2/3 groups agreed, consensus on the competency was arrived at through consultative discussion?

Response: We have rewritten and rephrased for clarity. Please see the seventh paragraph of the methodology.

---

## [Decision Letter · Decision Letter 1]

18 Jan 2021

Original Articles: Professional Competencies and Public Health Content in the Human Nutrition and Dietetics Degree Program: A Qualitative Study Based on Experts' Consensus

PONE-D-20-13991R1

Dear Dr. Vives-Cases,

We’re pleased to inform you that your manuscript has been judged scientifically suitable for publication and will be formally accepted for publication once it meets all outstanding technical requirements.

Kind regards,

Mauro Lombardo

Academic Editor

PLOS ONE

Additional Editor Comments (optional):

Reviewers' comments:

Reviewer's Responses to Questions

**Comments to the Author**

1. If the authors have adequately addressed your comments raised in a previous round of review and you feel that this manuscript is now acceptable for publication, you may indicate that here to bypass the “Comments to the Author” section, enter your conflict of interest statement in the “Confidential to Editor” section, and submit your "Accept" recommendation.

Reviewer #2: All comments have been addressed

Reviewer #3: All comments have been addressed

2. Is the manuscript technically sound, and do the data support the conclusions?

Reviewer #2: Yes

Reviewer #3: Yes

3. Has the statistical analysis been performed appropriately and rigorously? 

Reviewer #2: I Don't Know

Reviewer #3: N/A

4. Have the authors made all data underlying the findings in their manuscript fully available?

Reviewer #2: Yes

Reviewer #3: Yes

5. Is the manuscript presented in an intelligible fashion and written in standard English?

Reviewer #2: Yes

Reviewer #3: No

6. Review Comments to the Author

Reviewer #2: All comments have been implemented. I recommend accepting the manuscript.

All comments have been implemented. I recommend accepting the manuscript.

Reviewer #3: All the comments have been addressed. However, let me suggest few more aspects.

1. Regarding the description of the group (Methods section, line 158). I understand that all the participants had a PhD. Also, in the group were doctors (n=5). Were those “doctors” Physicians? If that is correct, it should state (as Physician) to make clearer that were 5 Physicians and all the participants were PhD or hold doctorates.

2. Include the word diet in line 99: “of professionals prepared to use a public health approach to address problems related to diet and nutrition.

3. Line 217: number three should be replaced by the number (to follow the way how is written before)

4. Although discussion mention all the competencies that Dietitian/Nutritionist should addressed, authors also mention that many of those are multidisciplinary nature of the field. Few lines are necessary to state why this human (Dietitian/Nutritionist) resource is the most suitable to accomplish those activities (lines 300-305).

7. PLOS authors have the option to publish the peer review history of their article (what does this mean?). If published, this will include your full peer review and any attached files.

Reviewer #2: No

Reviewer #3: No

---

## [Editor Report · Acceptance letter]

21 Jan 2021

PONE-D-20-13991R1 

Professional Competencies and Public Health Content in the Human Nutrition and Dietetics Degree Program: A QualitativeStudy Based on Experts' Consensus 

Dear Dr. Vives-Cases:

I'm pleased to inform you that your manuscript has been deemed suitable for publication in PLOS ONE. Congratulations! Your manuscript is now with our production department. 

Kind regards, 

on behalf of

Dr. Mauro Lombardo 

Academic Editor

PLOS ONE